

# Discovery of protein-based natural hydrogel from the girdle of the 'sea cockroach' *Chiton articulatus* (Chitonida: Chitonidae)

Emel Çakmak[1,2], Behlül Koc-Bilican[2,3],
Omar Hernando Avila-Poveda[4,5,6], Tuğçe Karaduman[2,3],
Demet Cansaran-Duman[7], Suzanne T. Williams[8] and Murat Kaya[2,3]

[1] Vegetable and Animal Production, Güzelyurt Vocational School, Aksaray University, Aksaray, Turkey
[2] Science and Technology Application and Research Center, ASUBTAM - Aksaray University, Aksaray, Turkey
[3] Molecular Biology and Genetics, Faculty of Science and Letters, Aksaray University, Aksaray, Turkey
[4] Facultad de Ciencias del Mar, Universidad Autónoma de Sinaloa, Mazatlán, Sinaloa, México
[5] Programa Investigadoras e Investigadores por Mexico, Consejo Nacional de Ciencia y Tecnología, Ciudad de México, México
[6] Proyecto Quitón del Pacífico Tropical Mexicano, Mazatlán, Sinaloa, México
[7] Biotechnology Institute, Ankara University, Ankara, Turkey
[8] Department of Life Sciences, Natural History Museum, London, Cromwell Road, United Kingdom

Corresponding author
Murat Kaya,
muratkaya3806@gmail.com

## ABSTRACT

Hydrogels are widely used materials in biomedical, pharmaceutical, cosmetic, and agricultural fields. However, these hydrogels are usually formed synthetically *via* a long and complicated process involving crosslinking natural polymers. Herein, we describe a natural hydrogel isolated using a 'gentle' acid treatment from the girdle of a chiton species (*Chiton articulatus*). This novel hydrogel is shown to have a proliferative effect on mouse fibroblast cells (cell line, L929). The swelling capacity of this natural hydrogel was recorded as approximately 1,200% in distilled water, which is within desired levels for hydrogels. Detailed characterizations reveal that the hydrogel consists predominantly (83.93%) of protein. Considering its non-toxicity, proliferative effect and swelling properties, this natural hydrogel is an important discovery for material sciences, with potential for further applications in industry. Whether the girdle has some hydrogel activity in the living animal is unknown, but we speculate that it may enable the animal to better survive extreme environmental conditions by preventing desiccation.

## INTRODUCTION

Hydrogels are three-dimensional hydrophilic networks that can absorb large amounts of water without dissolving (*Liao & Huang, 2020*; *Schiller & Lai, 2020*; *Varaprasad et al.,*

*2017*). This ability of hydrogels to take up large amounts of water by swelling makes them desirable materials for drug delivery (*Beninatto et al., 2019*; *Hoare & Kohane, 2008*), as a matrix for the growth of artificial organs, wound dressing, cancer treatments and tissue engineering (*Chao, Chen & Liu, 2020*; *Fan et al., 2019*; *Griffin et al., 2021*; *Ji et al., 2021*; *Zhong et al., 2020*), food coating, agriculture (*Klein & Poverenov, 2020*), antimicrobial delivery (*Jayaramudu et al., 2019*) and wastewater treatment (*Jing et al., 2013*). However, most commercially available hydrogels are formed synthetically from petroleum-based plastics or by combining natural biopolymers with crosslinking agents (*Fan et al., 2019*; *Zhong et al., 2020*).

These synthetic hydrogels are often preferred due to their high processability for customization, both in terms of their chemical composition and the ease with which mechanical properties can be adjusted, greater reproducibility, long term stability and the wide variety of easily accessible raw materials (*Samadian et al., 2020*; *Voorhaar & Hoogenboom, 2016*). However, excessive use of synthetic hydrogels can in some instances give rise to enormous health and environmental problems (*Liao & Huang, 2020*). In addition, the biocompatibility and non-toxicity of synthetic hydrogels cannot be guaranteed (*Gajendiran, Rhee & Kim, 2018*; *Samadian et al., 2020*). Commercially available 'natural' hydrogels (such as cellulose, chitosan, starch, alginate and gelatin) have some superior physicochemical and biological properties when compared with synthetic hydrogels (*Elvira et al., 2002*; *García-Astrain et al., 2016*; *Jiang & Kobayashi, 2017*; *Paukkonen et al., 2017*; *Zhang et al., 2019*). The most important of these are their sustainability, environmentally friendly composition, low immunogenicity, excellent biocompatibility and cytocompatibility, biodegradability, specific cellular responses, presence of antigens, cell proliferation controllability and 3D geometric structures (*Choi et al., 2019*; *Liao & Huang, 2020*; *Samadian et al., 2020*; *Shi et al., 2016*; *Zhong et al., 2020*). Although these hydrogels are described as 'natural', they are not naturally sourced, but are formed in a laboratory by crosslinking of natural polymers (*e.g.* cellulose and chitosan) after dissolution in a chemical solution. These processes can lead to poor mechanical properties and increased toxicity (*Jiang & Kobayashi, 2017*; *Paukkonen et al., 2017*). In addition, difficulties in obtaining sufficient quantities of natural starting polymers, the necessity for two or more steps in the production process and high production costs of the 'natural' crosslinked-polymer hydrogels further limit their utility (*Hoare & Kohane, 2008*; *Taylor & in het Panhuis, 2016*). It is extremely desirable to identify a completely natural hydrogel that would eliminate the need for toxic synthetic polymers and chemicals in their production. For this reason, marine organisms that are abundant in nature may serve as an alternative source to produce hydrogels with desirable properties (*Varaprasad et al., 2020*).

Herein, we report a completely natural hydrogel obtained directly from chiton girdle tissue with a 'gentle' acid treatment, and without the use of any cross-linkers or synthetic polymers. Chitons (Polyplacophora: Mollusca) are slow-moving, bilaterally symmetrical animals with eight calcareous plates overlying a dorsoventrally flattened body and a large muscular foot. Most species live on hard substrates and although they feed by grazing, seven different ecological feeding strategies have been identified (*Sigwart & Schwabe, 2017*;

*Sirenko, 2000*). Within the Polyplacophora, there are more than 922 recent and 368 fossils valid species worldwide (*Schwabe, 2005*). The edible chiton, *Chiton articulatus*, known locally as the 'sea cockroach' or 'dog's tongue', is a large species, occurring in high densities on the rocky intertidal splash zone along the Mexican Tropical Pacific (*Bullock, 1988*; *Reyes-Gómez, 2004*). It occurs most commonly in sites where animals are exposed to strong wave action, although their distribution is likely affected by human fishing efforts, which may reduce abundances in more easily accessible areas (*García-Ibáñez et al., 2013*; *Holguin-Quiñones & Michel-Morfín, 2002*).

The tissue used in this study is the integument tissue around the chiton body scleritome known as the girdle (*Schwabe, 2010*). Girdle morphology is modified in some chiton species, reflecting adaptations to different lifestyles; in some carnivorous species it is expanded and acts as a net to trap small prey items, other species have a slit in the girdle to facilitate excretion of waste (*Schwabe, 2010*). The chiton girdle is covered by a thick, organic, glycoproteinaceous cuticle (*Eernisse & Reynolds, 1994*) composed of a very thin, electron-dense, outer layer and a thick inner electron translucent layer composed of lamellae (*Checa, Vendrasco & Salas, 2017*). The girdle is often ornamented with calcareous and chitinous elements including scales, spines, spicules, needles, bristles and hairs that differ in appearance and occurrence among species (*Eernisse & Reynolds, 1994*). The appearance of armature provided by these small, mineralised scales has inspired the design of flexible, man-made armour (*Connors et al., 2019*), although their biological function for the animal is uncertain (*Leise & Cloney, 1982*).

During an attempt to isolate chitin from the girdle of *C. articulatus*, a new natural hydrogel was discovered, which we discuss in this study; however, no hydrogel formation was observed in the girdle of two other chiton species: *Plaxiphora aurata* (Mopaliidae, Mopaliinae) or *Rhyssoplax olivacea* (Chitonidae, Chitoninae). The properties of the sea cockroach hydrogel were determined using Fourier-transform infrared spectroscopy (FTIR), thermogravimetric analysis (TGA), X-ray diffraction analysis (XRD), scanning electron microscope (SEM) and energy-dispersive X-ray spectroscopy (EDS). In addition, 3-(4,5-Dimethylthiazol-2-yl)-2,5-Diphenyltetrazolium Bromide (MTT) assay and Real-Time, Quantitative Cell Analysis, xCELLigence system were used to test for cytotoxic effects.

## MATERIALS AND METHODS

### Sample collection and materials

Adult specimens of *Chiton articulatus* ($n$ = 80; 45 mm ≤ total length ≤ 80 mm) were collected from rocky intertidal shore at the northern limit of its geographical distribution, in Barras de Piaxtla, Sinaloa, Mexico (23°38′51.0″N 106°48′16.6″W), following collecting regulations established under Mexican law (NOM-126-SEMARNAT-2000). Individuals were relaxed by leaving them in a refrigerator at 7 °C for 30 min before dissection, following regulations for the humanitarian killing of animals as established under Mexican law (NOM-033-SAG/ZOO-2014). Subsequently, the soft tissues (foot, gills, gonads and viscera) were removed to allow the chiton scleritome (the eight interconnecting shelly plates) together with the girdle to be isolated (Fig. S1). Each chiton scleritome and girdle

were sun-dried for 2 days prior to further processing. Once dry, samples were stored away from further exposure to heat and light until the demineralization process.

## Protein-based hydrogel production

The scleritome and girdle were washed several times with distilled water to remove any possible dirt and particles. Shell plates were dissected from the body structure and the softened girdle structure was separated intact. Then to remove the calcareous shell layer on the surface of the girdle, specimens were treated with 0.5 M HCl (Sigma Aldrich, St. Louis, MO, USA) over 24 h. The integrity of the girdle structure was disrupted after the acid treatment, and pieces of girdle tissue were placed on dialysis membrane (Seamless Cellulose Tubing, size: 16/32, lot: 208001; Viskase Sales Corp., Chicago, IL, USA) and kept in distilled water with frequent changes of water over 3 days to ensure complete removal of the acid. Afterwards, the hydrogel samples were dried in an oven at 60 °C for 1 day. Although the same procedures were attempted, hydrogel formation was not observed in two other chiton species *Plaxiphora aurata* and *Rhyssoplax olivacea*.

## Characterization

Infrared spectra of hydrogel were obtained using a Perkin Elmer Spectrum FT-IR Spectrometer fitted with a Universal Attenuated Total Reflectance at 8 cm$^{-1}$ resolutions in the wavelength range of 600–4,000 cm$^{-1}$. This analysis, based on the principle of refracting X-rays in a characteristic order depending on the atomic sequences of each crystal phase, was carried out with Bruker AXS D8 Advance in the range of 40 kV, 30 mA, $2\theta$ at scanning range of 5–45°, with the final result based on the average of 64 scans to improve the signal-to-noise ratio. The percent crystallinity of the hydrogel sample was determined by using the intensity of the peaks obtained from XRD analysis. Crystallinity was calculated according to the following formula;

$$CrI_{110} = [(I_{110} - I_{am})/I_{110}] \times 100 \tag{1}$$

CrI = % crystallinity value, $I_{110}$ = maximum intensity value at $2\theta = 20°$, $I_{am}$ = maximum intensity value of the amorphous peak at $2\theta = 13°$.

The thermal stability of the hydrogel was determined using a Thermogravimetry/ Differential Thermal Analyzer (TGA Exstar-TG/DTA 7300 Instruments). The analysis was carried out under a nitrogen atmosphere at a heating rate of 10 °C/min in the range of 30–730 °C using a platinum crucible.

The surface morphology of the hydrogel was revealed by scanning electron microscopy (SEM) at 5 kV over a range of different magnifications (500×–30,000×). The swollen protein-based hydrogel sample sheets were freeze-dried for 24 h at −20 and −80 °C. To improve image quality, the material was gold-plated before scanning, using a Cressington sputter-coated 108 Auto. Energy dispersion spectrum (EDS) of protein-based hydrogel was measured at 20 kV and 5,000× magnification (EDAX-Octane Pro).

To determine the percentages of C, N, O and H elements in the structure of the protein-based hydrogel, high precision elemental analysis using SEM was conducted with a Thermo Flash 2000 microscope.

The Kjeldahl method was used to measure the nitrogen content of the dried hydrogel (1 g) following official methods (*Bradstreet, 1954*). After determining the total nitrogen content of the samples, crude protein content was calculated using a conversion factor of 6.25 to convert % nitrogen to % crude protein since most meat proteins characteristically contain 16% nitrogen (*Salo-väänänen & Koivistoinen, 1996*). Each sample was analysed in duplicate.

## Swelling study

The swelling behaviour of hydrogel samples was gauged by placing samples in distilled water and aqueous solutions of 0.9% NaCl, $MgCl_2$, $CaCl_2$ and $FeCl_3$ by weight using the gravimetric technique. Until swelling equilibrium was reached, dry and constant weight hydrogel (0.05 g) samples were kept in distilled water and salt solutions at room temperature. The swollen hydrogel was then placed on blotter paper and weighed after removing excess water from the surface. The swelling ratio was calculated according to the following equation;

$$\text{Swelling ratio } (\%) = \frac{Ws - Wd}{Wd} \times 100\% \tag{2}$$

where $wd$ is the dry sample weight; $ws$ is the weight of the swollen sample.

The water retention rate of the protein-based hydrogel was measured at different temperatures. Samples reached swelling equilibrium in distilled water were incubated in the oven at 30, 50 and 80 °C and weighed every hour for 12 h. Water retention rate (WR);

$$\text{WR } (\%) = \frac{Wt - Wd}{Ws - Wd} \times 100\% \tag{3}$$

where $wt$ is the weight of the sample at time t. Other arguments are the same as those defined earlier.

To evaluate the reusability of the protein-based hydrogel, the dried products (0.05 g) were suspended in distilled water (100 mL) at room temperature until reaching swelling equilibrium. The swollen samples were then weighed and the water holding capacity calculated according to the equation (Eq. (2)). Afterwards, the swollen sample was dehydrated in the oven at 60 °C until it reached a constant weight. For the next swelling experiment, an equal volume of distilled water was added to the recovered hydrogel sample. To determine the re-swelling capacity of the protein-based hydrogel, the same procedures were repeated five times and recorded.

## Determination of cytotoxic effects of the protein-based natural hydrogel obtained from the girdle of *C. articulatus*
### Cell culture

Mouse fibroblast cells (cell line L929) obtained from Sap Institute, Ankara, Turkey were cultured in Dulbecco's Modified Eagle's Medium (DMEM, Biological Industries®, Cromwell, CT, USA) supplemented with 10% Fetal Bovine Serum (FBS; Biowest®, Riverside, MO, USA), 1% Penicillin-Streptomycin solution (Biowest®, Riverside, MO, USA) at 37 °C, 5% $CO_2$ humidified incubator and checked daily by using an inverted

microscope (Leica DM IL LED; Leica Microsystems, Wetzlar, Germany). The cells were passaged at 80–85% confluence.

### Cytotoxicity assay

The cytotoxic effect of hydrogel was evaluated using an MTT analysis, with L929 cells seeded in 96-well plates at $1 \times 10^4$ cells per well in 100 µl of complete DMEM. After 24 h of incubation, three different amounts of sterilized blown protein-based hydrogel (1, 2, and 4 mg) were placed in contact with the cells and then incubated from 24 h to 72 h at 37 °C in a humidified atmosphere of 5% $CO_2$. Following the incubation period, sterilized hydrogel pieces were removed and MTT (0.5 mg/mL) solution was added to each well and further incubated for 4 h at 37 °C. Afterward, the reaction mixture was removed from each well, replaced by 100 µl of dimethyl sulfoxide (DMSO) solution, and the optical density (OD) was measured at 492 nm by ELISA reader (ChroMate®; Awareness Technology, Inc., Palm City, FL, USA). The measured absorbances were directly proportional to the number of living cells and the viability in hydrogel-free control groups was calibrated to be 100%. Cell viability in the treatment groups was compared with the control group and calculated by the following equation;

$$\%\text{Viable cells} = ((A_1 - A_0)/(A_2 - A_0)) \times 100\% \tag{4}$$

where $A_0$ is the absorbance of the blank (medium without mouse cells), $A_1$ absorbance of 1, 2 and 4 mg hydrogels, $A_2$ is the absorbance of the control (cells in solution, grown without any hydrogel).

Data analysis of MTT results was performed using GraphPad Prism software version 8 (GraphPad Software®, San Diego, CA, USA). The data represent the mean ± SD (standard deviation). Statistical differences were evaluated by two-way ANOVA with a Bonferroni correction (95% confidence interval). Values with $p < 0.05$ were regarded as statistically significant.

An xCELLigence Real-Time Cell Analysis instruments (RTCA; ACEA Biosciences, Roche, Germany) system was also used for real-time and label-free online monitoring of cytotoxicity (*Stefanowicz-Hajduk & Ochocka, 2020*). The $1 \times 10^4$ cells/well were seeded in a 16-well e-plate (RTCA; ACEA Biosciences, Roche, Germany) and then the cell index (CI) was measured every 15 min for 96 h. After the growth period of cells, the hydrogel pieces (1 and 2 mg) were added to 16-wells in the e-plate. The CI value of the control (cell with DMEM) and hydrogel treated samples were graphed by using the xCELLigence RTCA software version 2.1.0.

## RESULTS

### A novel hydrogel

As a serendipitous result of an attempt to characterize chitin in girdle tissue from the 'sea cockroach', *Chiton articulatus*, we discovered a new, natural hydrogel, which can be produced in just a few, simple steps and without the need for expensive equipment or toxic chemicals. Figure 1 describes the process of converting the chiton girdle to a protein-based hydrogel.
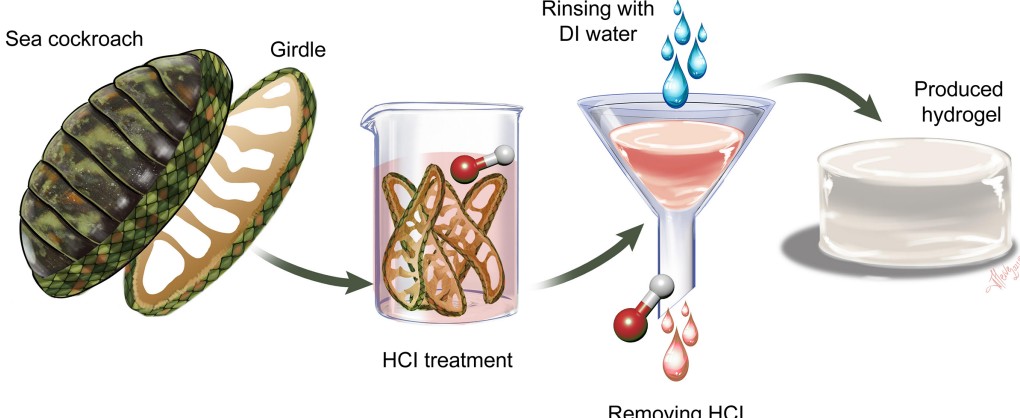

**Figure 1 Process for producing hydrogel from chiton girdle.** First, the girdle structure is separated from the chiton body scleritome by dissection. The separated girdles are subjected to 0.5 M HCl. Then the samples are filtered by rinsing with distilled water and the natural hydrogel is produced.

## Characterization

The FTIR spectrum of *C. articulatus* hydrogel isolated from the girdle structure is shown in Fig. 2A. For the hydrogel isolated from *C. articulatus*, Amide I, Amide II and Amide III bands were recorded at 1,625 cm$^{-1}$, 1,537 cm$^{-1}$ and 1,235 cm$^{-1}$, respectively. These recorded peaks suggest that the hydrogel is protein-based. Other peaks recorded during FTIR analysis are given in Table S1.

The crystal structure of the protein-based hydrogel isolated from *C. articulatus* was revealed by XRD measurement (Fig. 2B). The XRD peaks of the hydrogel obtained from the girdle structure of *C. articulatus* were found to be compatible with previous protein studies and the crystallinity was calculated as 84.9% (*Alashwal, Gupta & Husain, 2019*; *Ki et al., 2007*; *Meng et al., 2012*).

Results of Thermogravimetry (TGA) and Differential Thermal analyses (DTA) of the protein-based hydrogel isolated from *C. articulatus* are given in Fig. 2C. The peak clearly observed in the figure is due to degradation of the protein. The maximum decomposition temperature value was observed as 305.9 °C and 77.5% mass loss was recorded in this decomposition.

Surface morphology and pore structure of hydrogels are the main characteristics used to determine their potential applications (*Bashir et al., 2020*). Therefore, to reveal the surface properties of the protein-based hydrogel, SEM analysis was conducted after drying samples using two different lyophilization techniques, and the surface morphologies are presented in Figs. 2D and 2E. The pore sizes range in size, with the largest reaching 100 μm.

The elemental composition of the hydrogel was determined to be 11.63% nitrogen, 44.45% carbon, 6.46% hydrogen and 0.68% sulfur (Fig. 2F). In the EDS analysis, C, N, H and S as well as very small amounts of Na, Mg and Cl were detected in protein-based hydrogel. The result of Kjeldahl analysis revealed that the chiton hydrogel contains as 83.93 g/100 g protein by dry weight.

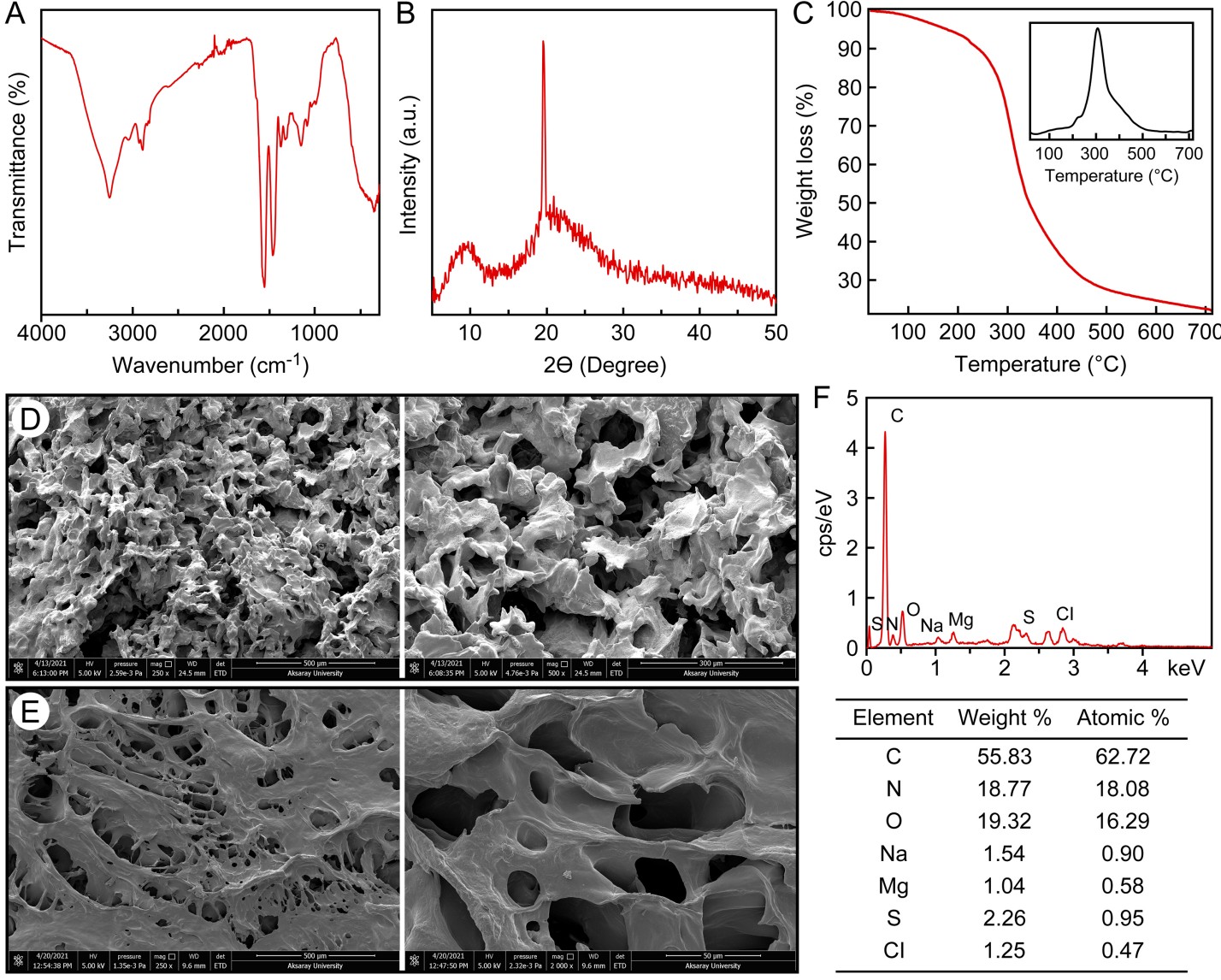

**Figure 2** Characterization of the protein-based hydrogel by (A) FT-IR, (B) XRD, (C) TGA, (D) SEM, −20 °C freeze-dried, (E) SEM, −80 °C freeze-dried, (F) EDS and elemental analysis.                         

## Swelling study

The swelling rate (%) of the protein-based hydrogel over time in distilled water and saline solution types is given in Fig. 3A. It clearly shows a similar swelling behaviour when placed in distilled water, and 0.9% by weight aqueous solutions of $FeCl_3$, $MgCl_2$, $CaCl_2$ and NaCl, to that exhibited by other hydrogel samples. In the saline solutions, it is evident that the swelling of the hydrogel samples is dramatically reduced compared to its uptake of distilled water. The maximum water absorption values of the samples in distilled water, $FeCl_3$, $MgCl_2$, $CaCl_2$ and NaCl were 1,200%, 450%, 400%, 350%, 325%, respectively. Figures 3D–3F shows the morphology after swelling.

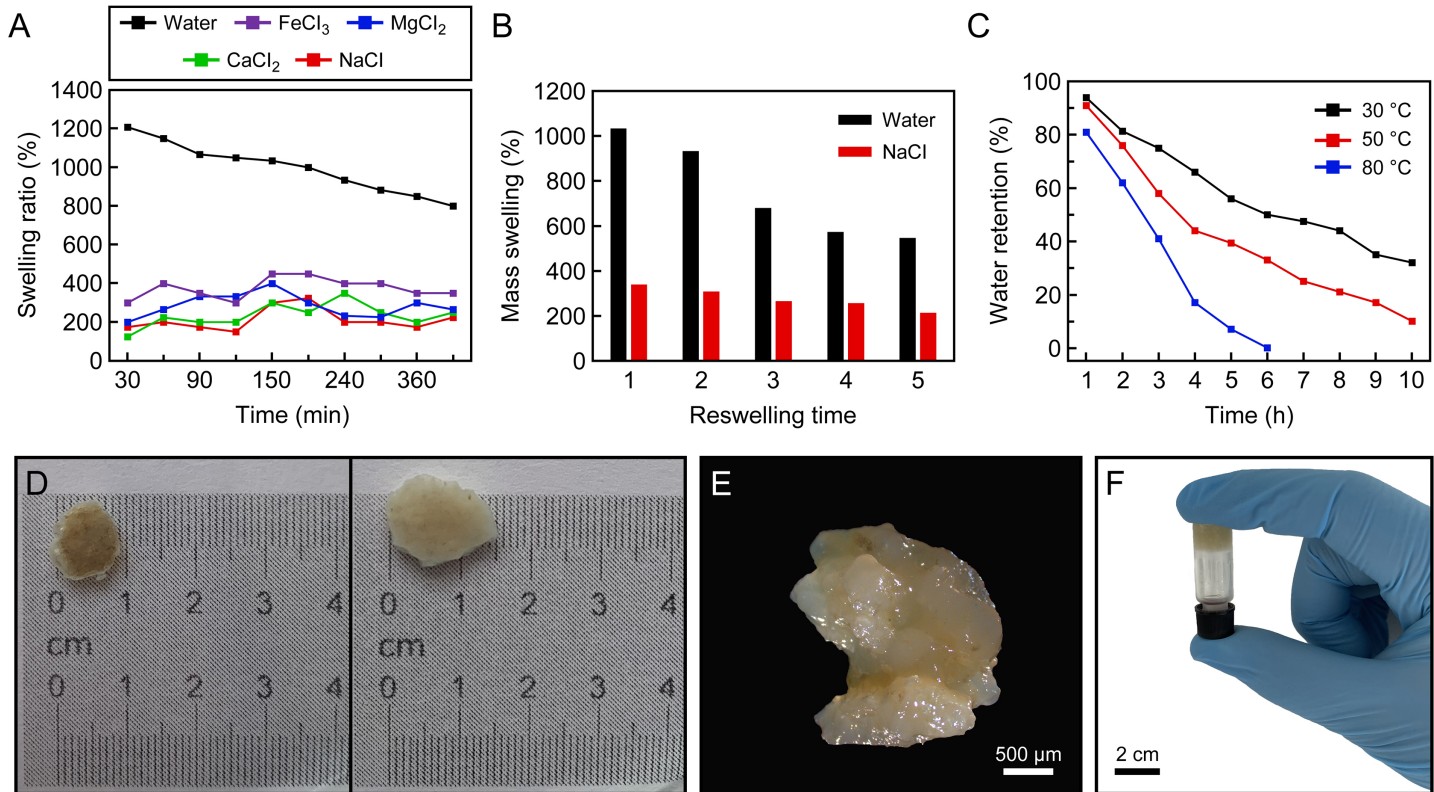

**Figure 3 Swelling ratio.** (A) Swelling ratio of hydrogels in distilled water and different salt solutions (0.9% by weight): NaCl, CaCl₂, MgCl₂ and FeCl₃, (B) reswelling capability of protein-based hydrogel, (C) effect of temperature on water-retention (%) of protein-based hydrogel, (D) the natural hydrogel swells and keeps their structural integrity, (E) the image of swelled hydrogel taken by light microscopy, (F) an inverted test tube holds the protein-based hydrogel.

The water absorbency of the protein-based hydrogel in distilled water and 0.9 wt% NaCl solution as a function of the number of reswelling time was shown in Fig. 3B. Looking at the swelling capacity of five cycles in distilled water and 0.9% NaCl solution, it was seen that the reswelling capacity of the hydrogel samples gradually decreases as the number of reswelling cycles increases (Fig. 3B). It can be seen that protein-based hydrogel samples were still able to maintain high water content even after five cycles of swelling and drying: approximately 546% in distilled water and 213% in 0.9 wt% NaCl solution.

Figure 3C showed the water-retention capacity of the protein-based hydrogel at three temperatures (30, 50 and 80 °C). The absorbency of the samples kept at 32 and 10 wt% after heating at 30 and 50 °C for 10 h, respectively, and 0.8% at 80 °C for 6 h. In addition, it is understood from Fig. 3C that the water absorbed in the hydrogel can be released with increasing temperature.

## Cytotoxicity assay

To assess the level of biocompatibility of the *C. articulatus* hydrogel for medical uses, different amounts of hydrogel were applied to the L929 mouse fibroblast cells (Figs. 4A and 4B). Cell viability according to MTT assay results for control (without any hydrogel), 1, 2 and 4 mg hydrogel-treated wells at 24 h were determined as 100%, 124.2%, 98.15%,

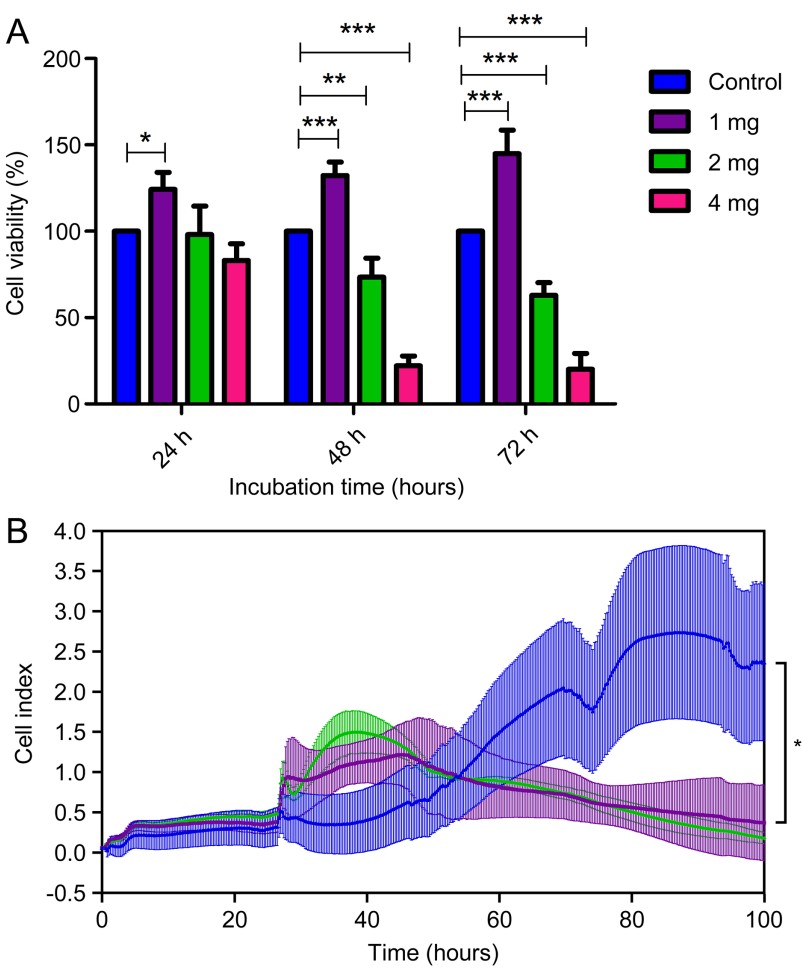

**Figure 4** (A) Cell viability assays of mouse fibroblast L929 cells after being cultured with chiton hydrogel (1, 2, and 4 mg) for 24 h, 48 h and 72 h (bars represent mean cell viability ± SD; $n$ = 3 statistical difference: $^*p < 0.05$, $^{**}p < 0.01$, $^{***}p < 0.001$), (B) real-time analysis.

and 83.01% respectively. Viability changed to 100%, 132.3%, 73.46%, and 22.15% at the end of 48 h, and 100%, 144.9%, 62.99%, and 20.13% after 72 h. Accordingly, the viability of the cells treated with 1 mg hydrogel was significantly increased in all three incubation times (24, 48 and 72 h) compared to the control ($p < 0.05$). In addition, a significant decrease in cell viability was observed at 48 and 72 h in 2 and 4 mg hydrogel applications ($p < 0.05$).

The MTT assay results were then validated with a real-time experiment examining cell viability in the xCELLigence system. The xCELLigence system measures cellular changes under real-time conditions as an electric impedance of the e-plate. We treated L929 non-cancerous mouse fibroblast cells with the 1 and 2 mg concentrations of hydrogel by xCELLigence system. The real-time analysis demonstrated a gradual increase of the cell index with 1 and 2 mg concentrations of hydrogel for 25 h on L929. Moreover, 1 mg hydrogel showed a proliferative effect on L929 cells until the 91st h (67 h after the addition of hydrogel) ($p < 0.05$). MTT and xCELLigence assay results are given in Fig. 4B. In the

light of these important results, the absence of cytotoxic effects of hydrogel-based biological materials suggests *C. articulatus* hydrogel shows a significant potential for biotechnological and biomedical applications.

## DISCUSSION

### Characterization

Amide I (1,600–1,700 cm$^{-1}$), Amide II (1,504–1,582 cm$^{-1}$) and Amide III (1,200–1,300 cm$^{-1}$) absorption bands are characteristic for protein-based materials (*Castrillón-Martínez et al., 2017*). The wide peak recorded at 3.275 cm$^{-1}$ in the spectrum is due to the vibration of the O–H group connected by intermolecular hydrogen bonding of the absorbed water (*Pourjavadi et al., 2006*). These peaks are consistent with the results of the FTIR analysis of hydrogels prepared from collagen and silk fibroin proteins in the literature (*Montalbano et al., 2018*; *Motta et al., 2004*).

According to the XRD analysis, the peaks clearly observed for the hydrogel are 9.7 and 19.6°. It has been reported in the literature that the peaks recorded for other structural proteins such as collagen, keratin and silk are in the range of 7.8–10.7° and 19.6–21.8° (*Alashwal, Gupta & Husain, 2019*; *Ki et al., 2007*; *Meng et al., 2012*).

The maximum decomposition temperature recorded for protein-based hydrogel isolated from the girdle structure of *C. articulatus* was 305.9 °C, which was slightly lower than decomposition temperatures reported for other structural proteins (collagen, keratin, fibroin, etc.) in the literature (*Chomachayi et al., 2020*; *Kakkar et al., 2014*; *Mekonnen, Ragothaman & Palanisamy, 2017*). The maximum decomposition temperatures recorded for hemicellulose-based hydrogels produced in a previous study were 300–375 °C (*Guan et al., 2014*). In another study, the degradation temperature for starch based hydrogel was recorded as 317–322 °C (*Vakili & Rahneshin, 2013*). Synthetic hydrogels have higher thermal stability than biopolymer-based composite hydrogels. In an earlier study, decomposition of hydrogel backbone for surfmer-co-poly acrylates crosslinked hydrogels occurred 315–430 °C (*El-Hoshoudy et al., 2019*). In another study, in the thermogram recorded for hybrid polyacrylamide hydrogels, the degradation of cross-linked polymers occurred at 405–560 °C, while the crosslinker decomposed at 560–794 °C. In the same study, it was noted that the decomposition temperature increased as the crosslinker ratio increased (*Nadtoka et al., 2018*). More than one degradation step was recorded in all of the TGA analyzes performed on hydrogels in the literature (*El-Hoshoudy et al., 2019*; *Guan et al., 2014*; *Kumar, Kaith & Mittal, 2012*; *Lessa, Nunes & Fajardo, 2018*; *Nadtoka et al., 2018*; *Peng et al., 2015*; *Rusu et al., 2015*; *Vakili & Rahneshin, 2013*). These results reveal that synthetic hydrogels are composed of many different components, including crosslinkers. In the present study, the presence of a single degradation peak for the protein-based hydrogel isolated from *C. articulatus*, suggests that it consists of almost entirely of protein. This is confirmed by the result of the Kjeldahl analysis, which revealed that the chiton hydrogel contains 83.93% protein by dry weight.

Similarly high concentrations of crude proteins have been observed in fish protein-based hydrogel (*Hwang & Damodaran, 1997*) and soy protein-based hydrogels, which are typically purified for industrial and research fields (*Abaee, Mohammadian & Jafari, 2017*).

As seen in previous studies of natural hydrogels (*Polat, Duman & Tunç, 2020*; *Qi et al., 2020*; *Zhang et al., 2020*), a porous and rough morphology was observed in both protein-based hydrogel samples, however the first drying technique (Fig. 2D) makes the hydrogel more porous than the second drying technique (Fig. 2E). The pore sizes range in size, with the largest reaching 100 μm. Surface morphologies and pore structures of hydrogels vary considerably according to the drying technique and the solvents used (*Kumar & Han, 2017*; *Laftah, Hashim & Ibrahim, 2011*; *Mahinroosta et al., 2018*; *Ullah et al., 2015*). Hence the effects of using other drying techniques and solvents on *C. articulatus* hydrogel need to be investigated in further studies.

Our elemental analysis values are similar to those reported for some other structural proteins such as keratin, sericin and fibroin (*Jena et al., 2018a*; *Jena et al., 2018b*; *Shavandi et al., 2017*; *Xia & Lu, 2008*). It has been reported in the literature that Mg and Na elements are associated with calcium phosphates (*León-Mancilla et al., 2016*). These elements are thought to be present because the upper layer of the girdle structure, from which the hydrogel is obtained, is completely covered with shell plates (*Connors et al., 2019*). In a previous study, C, H, N, O and S were similarly detected in chitosan, silk fibroin and egg shell membrane hydrogels (*Adali, Kalkan & Karimizarandi, 2019*). In addition to these, C, O, N and H were also detected in the synthetically produced bioconjugated graphene oxide hydrogel (*Soleimani, Tehrani & Adeli, 2018*). In another study, high levels of P as well as C and H were recorded in the synthetically produced phosphate-containing hydrogel (*Wang et al., 2003*). This result shows that these detected elements in the synthetic hydrogels are related to the plastic polymeric materials added to the composition of the hydrogel.

## Swelling study

The ability of hydrogels to take up large amounts of liquids means they have the potential for the sustainable release of absorbed molecules, and this capacity is of primary importance in many practical applications such as water release systems in agriculture (*Hosseinzadeh, 2013*). Properties such as charge of cations and salt concentration greatly affect the swelling behaviour of hydrogels (*Tanan, Panichpakdee & Saengsuwan, 2019*). Looking at the literature to compare the swelling capacities of synthetic and natural hydrogels, in the study of *Tanan, Panichpakdee & Saengsuwan (2019)*, it was shown that the maximum swelling capacity in water was determined as 794% in biodegradable hydrogel based on 'natural' polymers. However, in synthetic hydrogels, the percentage of maximum swelling ratio of the hydrogel has been shown to be up to 2,000% the dry mass (*Kiran, Krishnamoorthi & Kumar, 2019*). The swelling rate of hydrogels in salt solutions is thought to depend not only on salt concentration but also on ionic charge. In the study of *Tanan, Panichpakdee & Saengsuwan (2019)*, it was clearly shown that as the charge of cations increases, the swelling capacity decreases accordingly, but in our study, on the contrary, swelling after exposure to $FeCl_3$, $MgCl_2$ and $CaCl_2$ solutions was greater than that observed after exposure to NaCl. Similar results have been reported in the literature, with the swelling capacity of some hydrogels increasing with increasing charge density and decreasing salt concentration (*Chang et al., 2011*; *Liu, Tong & Hu, 1995*).

However, it has been shown in many studies that the swelling ratio of cations decreases as the salt concentration and value increase (*Li et al., 2017*; *Namazi, Hasani & Yadollahi, 2019*; *Wang et al., 2018*). In addition, it has been shown that the absorbance of nanocomposite hydrogels in salt solutions is relatively higher than that of pure hydrogel (*Namazi, Hasani & Yadollahi, 2019*). Nevertheless, the swelling rate of hydrogel samples also differs in different salt solutions (*Chang et al., 2011*). Similar to this study, *Chang et al. (2011)* clearly showed that the swelling ratio was significantly decreased with increasing ionic charge in some hydrogels, whereas other hydrogels showed minimal swelling in $CaCl_2$ instead of $FeCl_3$. This proves that hydrogels have smart swelling behaviors in aqueous solutions such as NaCl, $CaCl_2$ and $FeCl_3$.

The reswelling capacity is one of the most important characteristics of hydrogels for the application as a superabsorbent in practice, which shows the stable water absorption ability and water retention (*Li et al., 2012*; *Tanan, Panichpakdee & Saengsuwan, 2019*). The reason why the reswelling capacity of hydrogel samples gradually decreases as the number of re-swelling cycles increases is probably due to damage in polymeric network structures (*Tanan, Panichpakdee & Saengsuwan, 2019*), which may affect the water holding capacity of the hydrogel. But at the same time, it can be seen that protein-based hydrogel samples are still able to retain a high-water content, even after five cycles of swelling and drying. Judging by our results, the *C. articulatus* hydrogel will prove to be useful as a recyclable and reusable superabsorbent material, as a certain degree of water absorption ability is conserved, even after repeated reswelling cycles.

The water-retention capacity can be determined by the interaction of H bond and van der Waals forces between water molecules and hydrogels (*Vudjung et al., 2014*; *Wen et al., 2016*). Our results demonstrated that the water-retention capacity of the chiton protein-based hydrogel decreases almost linearly with the increase in temperature and time. *Lv, Wu & Shen (2019)* proved that super-absorbent hydrogels (SAH) show poor water-retention capacity when temperature rises. Obviously, the product has extensive properties that can be useful in many areas, primarily in agricultural applications where it could be used to retain moisture and provide nutrients to plants.

## Cytotoxicity assay

Cytotoxicity tests are important assays to determine the biocompatibility of materials to be used in medical applications (*Assad & Jackson, 2019*) (*e.g.* as scaffolds for drug delivery (*Motta et al., 2004*)). In this study, cytotoxicity analyzes were conducted using different amounts of a novel chiton protein hydrogel, which was obtained without the use of toxic chemicals that could adversely affect natural molecules in the biological structure. Traditional MTT analysis and xCELLigence system showed that the chiton protein-based hydrogel, used at an appropriate level (1 mg), is not only non-toxic, but has a proliferative effect on cells. Additionally, we predict that the hydrogel in this study may show a further increase effect on cell proliferation with the protein-based bioactive components in its structure when used at an appropriate doses and times. The observed time-dependent increase in cytotoxicity in cells with high hydrogel application (for 2 and 4 mg) was

associated with the increase in swelling capacity over time and, accordingly, with inadequate media and gas transportation to the cells.

## Biological function

A natural hydrogel, that does not require laboratory polymerisation, was isolated in this study from the girdle of *Chiton articulatus* (Chitonidae, Chitoninae) (Fig. 1). However, hydrogel was not found in two other chiton species examined: *Plaxiphora aurata* (Mopaliidae, Mopaliinae) and *Rhyssoplax olivacea* (Chitonidae, Chitoninae). The fact that hydrogel does not occur in all chiton species examined suggests that the tissue and structures responsible for hydrogel may be a species or clade-specific adaptation. Organisms, like *C. articulatus*, that live in the tropical, rocky intertidal splash zone must tolerate fluctuating and at times extremely high temperatures, levels of ultraviolet radiation and salinity. The ability to thrive in such extreme conditions relies on adaptations that provide protection against or a means to overcome these challenges. For instance, some studies have shown significant differences in levels of heat shock proteins in chiton species *Katharina tunicata* and *Chaetopleura angulata* sampled in different seasons (*Burnaford, 2004*; *Madeira et al., 2017*). The chiton *Mopalia mucosa* has been shown to undertake whole-animal volume regulation as evidenced by weight gain to control osmolarity (*Leise & Cloney, 1982*; *Moran & Tullis, 1980*), and in some species, girdle elements, such as overlapping scales and hairs that entrap mud and detritus, may help prevent desiccation during low tides (*Leise & Cloney, 1982*; *Moran & Tullis, 1980*).

Whether the girdle in *C. articulatus* retains in the living animal some of the hydrogel characteristics observed in this study is unknown. We speculate that it may play a biological role, enabling the animal to better survive the extreme environmental conditions it experiences in the rocky intertidal (*Flores-Garza et al., 2011*) by preventing desiccation and helping to regulate osmolarity of cells. In fact, in the field, *C. articulatus* have been observed with the girdle subtly swollen and detached from the body scleritome, revealing a part of each plate (sclerite or valve); this in comparison between the groups of chitons observed (O. H. Avila-Poveda, 2016, personal observation). Furthermore, it has been suggested that changes to the tissue volume of the foot would affect an animal's ability to attach firmly to its substrate (*McGill, 1975*), and our finding that tissues associated with the girdle, but not the foot, produce hydrogel is consistent with this idea. Exactly how the hydrogel would function in the animal is unknown and is worthy of future study.

## CONCLUSION

The biological properties of the novel *C. articulatus* hydrogel isolated in this study make it suitable for use as an industrial, three-dimensional scaffold material due to its behaviour and high biocompatibility with mouse fibroblast L929 cell growth. Further biological analyses, however, in particular more biocompatibility tests using different cell lines, are needed to determine its potential for use in biomedical applications such as biomedical engineering and regenerative medicine. Of relevance when considering the feasibility of using a chiton protein-based hydrogel for such applications, *C. articulatus* is currently being considered as a target for aquaculture (*Avila-Poveda, 2020*). Although reasonably

abundant in nature, farmed animals could provide a steady source of raw materials and offers the potential for selective breeding or genetic modification to change or improve hydrogel properties. Although we did not find hydrogels in two other species, surveys of other chiton species may also identify other sources for further novel hydrogels.

# ACKNOWLEDGEMENTS

O. H. Avila-Poveda gratefully acknowledges the participation of Camilla Guillen, Quetzalli Yasu Abadia-Chanona, Miranda Guadalupe Prado-Padilla, Mario Arath Lizarraga-Sanchez and Brenda Paola Ramirez-Santana, members of the project 'Quiton del Pacifico tropical mexicano', for their assistance with sampling and dissection. O. H. Avila-Poveda is a CONACYT Research Fellow hosted by the Universidad Autonoma de Sinaloa's Facultad de Ciencias del Mar (project no. 2137) in the research group 'Manejo de Recursos Pesqueros UAS-CA-132, UAS-FACIMAR'. Special thanks to Dr. İsmail Bilican for his contribution to the drawings.

## Funding

The authors received no funding for this work.

## Competing Interests

Suzanne T. Williams is an Academic Editor for PeerJ.

## Author Contributions

- Emel Çakmak conceived and designed the experiments, performed the experiments, analyzed the data, prepared figures and/or tables, and approved the final draft.
- Behlül Koc-Bilican performed the experiments, analyzed the data, prepared figures and/or tables, and approved the final draft.
- Omar Hernando Avila-Poveda performed the experiments, analyzed the data, authored or reviewed drafts of the paper, sample collection, and approved the final draft.
- Tuğçe Karaduman performed the experiments, analyzed the data, prepared figures and/or tables, and approved the final draft.
- Demet Cansaran-Duman performed the experiments, analyzed the data, prepared figures and/or tables, and approved the final draft.
- Suzanne T. Williams performed the experiments, analyzed the data, authored or reviewed drafts of the paper, and approved the final draft.
- Murat Kaya conceived and designed the experiments, performed the experiments, analyzed the data, authored or reviewed drafts of the paper, and approved the final draft.

## Field Study Permissions

The following information was supplied relating to field study approvals (*i.e.*, approving body and any reference numbers):

Specimens were collected under research permit PPF/DGOPA-130/15 and PPF/DGOPA-110/21 granted to O.H. Avila-Poveda by SAGARPA (currently Secretaria de

Agricultura y Desarrollo Rural 'SADER') through Comision Nacional de Acuacultura y Pesca 'CONAPESCA'.

Several adult specimens of *Chiton articulatus* were collected ($n$ = 80; 45 mm ≤ total length ≤ 80 mm), from rocky intertidal shore at the northern limit of its geographical distribution, in Barras de Piaxtla, Sinaloa, Mexico (23°38′51.0″N 106°48′16.6″W), following collecting regulations established under Mexican law (NOM-126-SEMARNAT-2000). Individuals were relaxed by leaving them in a refrigerator at 7 °C for 30 min before dissection, following regulations for the humanitarian killing of animals as established under Mexican law (NOM-033-SAG/ZOO-2014).

## Data Availability

The raw data is available in the Supplemental Files.

## Supplemental Information

Supplemental information for this article can be found online at http://dx.doi.org/10.7717/peerj.13386#supplemental-information.

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
