# Peer review of "Discovery of protein-based natural hydrogel from the girdle of the ‘sea cockroach’ Chiton articulatus (Chitonida: Chitonidae)"

_PeerJ, doi:10.7717/peerj.13386_

## Round 0.1 · original submission · Minor Revisions

· Academic Editor

Minor Revisions

Please address the concerns of both reviewers and amend your manuscript accordingly.

Reviewer 1 ·

Basic reporting

In this article Cakmak et al., report their discovery of a protein-based natural hydrogel from the girdle of a chiton species, Chiton articulatus. Further, they analyze the chiton hydrogel to study the thermal stability, surface morphology, chemical composition, swelling behavior and cytotoxicity. This chiton hydrogel could be a good alternative to the synthetic hydrogels and the so called ‘natural hydrogels’ due to their potential for being environmentally safe , non-toxic and sustainable. While a good amount of characterization has been performed it will be valuable to report the mechanical properties of the chiton hydrogel especially because the other ‘natural hydrogels’ have poor mechanical properties. In addition to this there are other minor issues such as text being difficult to read at some parts.


1) Describe fig 3F further.

2) Make sure the scientific names of the organisms are italicized. For example in line 199.

3) The last sentence in the abstract ends as “….and helping to regulate”. It would be useful if you the authors can specify what process they speculate is being regulated or revise the sentence.

4) Raw data files need to be titled and annotated clearly. It is difficult to understand them in their current form.

5) In figures and text, the terms chiton hydrogel and protein-based hydrogel seem to be used interchangeably. The authors should be consistent and use one term throughout the text. This would help with the flow of the article.

6) In line 321, the biological function section could be removed and the text added to the discussion.

7) In line 255, “previous protein studies” are mentioned. Please elaborate on how the results from this study are compatible with these previous protein studies and cite them.

8) Typo in line 264, Fig.S instead of Figs.

9) Revise the sentences 269 and 270. Their purpose is not clear. I assume the idea is to indicate that protein concentrations in the chiton hydrogels are higher than other hydrogels.

10) Rephrase sentences 307 and 308. As MTT assay is already mentioned in the preceding paragraph you could simply state “The MTT assay results were then validated with a real-time experiment examining cell viability…….”

11) In line 70, it is mentioned that ‘natural’ hydrogels are non-toxic but in line 77 increased toxicity is noted as one of the disadvantages of these ‘natural’ hydrogels. This is a bit confusing. It would be useful to explain or revise these statements further.

Experimental design

1) The paper refers to the so called ‘natural’ hydrogels which are prepared in the lab by crosslinking of natural polymers as having poor mechanical properties. It would be valuable if you can test and present the mechanical properties of the chiton hydrogel.

Validity of the findings

1) In line 339, mention the maximum decomposition temperature values and explain the issues if any related to this slightly lower decomposition temperature.

2) For the MTT assay results in figure 4, report the standard deviations rather than standard error of the mean. Standard deviation shows the variability from the individual data values to the mean which is appropriate here.

3) In paragraph Cytotoxicity assay, line 427, expand more on how chitin hydrogel can be used in biomedical field and the limitations of it based on the toxicity observed at higher concentrations.

4) In lines 390, 391 and 392, mention the maximum swelling capacity either as percentages or times. In addition, it would be useful for the reader to note the maximum swelling capacity of chitin hydrogel in this context.

5) Can you address if any difficulties in obtaining enough of this chitin hydrogel for different applications and possible environmental impact?

6) In lines 378 and 379, the purposes of these sentences without any reference to the chiton or protein-hydrogel is not clear. Revising them would help the reader to understand it better.

·

Basic reporting

Emel et al., developed protein-based hydrogels from the girdle of the sea cockroach.

Experimental design

The selected topic was very interesting and the authors presented it in a good manner way.

Validity of the findings

The selected topic was very interesting

Additional comments

Minor comments:
• The extracted material needs to compare with standard materials.
• Abstract: use this sentence “predominantly 83.93% of protein”
• Remove these “Chiton articulatus; Chitonidae”, if there are synonyms
• Add the following reference at Line 49: 10.1016/j.msec.2017.05.096; line 53: 10.1016/j.ijbiomac.2019.01.145; line 84: 10.1016/j.molliq.2019.112087 with explanation

---

## Round 0.2 · accepted · Accept

· Academic Editor

Accept

In my view, all the issues pointed out by the reviewers were adequately addressed and the manuscript was revised accordingly.